META-RESEARCH ARTICLE

# The methodological quality of 176,620 randomized controlled trials published between 1966 and 2018 reveals a positive trend but also an urgent need for improvement

Christiaan H. Vinkers[1]*, Herm J. Lamberink[2], Joeri K. Tijdink[3], Pauline Heus[4], Lex Bouter[5], Paul Glasziou[6], David Moher[7], Johanna A. Damen[4], Lotty Hooft[4], Willem M. Otte[8]

1 Department of Psychiatry and Department of Anatomy and Neurosciences, Amsterdam UMC, Amsterdam, the Netherlands, 2 Department of Child Neurology, UMC Utrecht Brain Center, University Medical Center Utrecht, and Utrecht University, Utrecht, the Netherlands, 3 Department of Ethics, Law and Humanities, Amsterdam UMC, and Department of Philosophy, Vrije Universiteit Amsterdam, Amsterdam, the Netherlands, 4 Cochrane Netherlands, Julius Center for Health Sciences and Primary Care, University Medical Center Utrecht, Utrecht University, Utrecht, the Netherlands, 5 Department of Epidemiology and Data Science, Amsterdam UMC, and Department of Philosophy, Vrije Universiteit Amsterdam, Amsterdam, the Netherlands, 6 Institute for Evidence-Based Healthcare, Bond University, Gold Coast, Australia, 7 Centre for Journalology, Clinical Epidemiology Program, The Ottawa Hospital Research Institute, Ottawa, Canada, 8 Department of Child Neurology, UMC Utrecht Brain Center, University Medical Center Utrecht, and Utrecht University, Utrecht, the Netherlands

* c.vinkers@amsterdammumc.nl

## Abstract

Many randomized controlled trials (RCTs) are biased and difficult to reproduce due to methodological flaws and poor reporting. There is increasing attention for responsible research practices and implementation of reporting guidelines, but whether these efforts have improved the methodological quality of RCTs (e.g., lower risk of bias) is unknown. We, therefore, mapped risk-of-bias trends over time in RCT publications in relation to journal and author characteristics. Meta-information of 176,620 RCTs published between 1966 and 2018 was extracted. The risk-of-bias probability (random sequence generation, allocation concealment, blinding of patients/personnel, and blinding of outcome assessment) was assessed using a risk-of-bias machine learning tool. This tool was simultaneously validated using 63,327 human risk-of-bias assessments obtained from 17,394 RCTs evaluated in the Cochrane Database of Systematic Reviews (CDSR). Moreover, RCT registration and CONSORT Statement reporting were assessed using automated searches. Publication characteristics included the number of authors, journal impact factor (JIF), and medical discipline. The annual number of published RCTs substantially increased over 4 decades, accompanied by increases in authors (5.2 to 7.8) and institutions (2.9 to 4.8). The risk of bias remained present in most RCTs but decreased over time for allocation concealment (63% to 51%), random sequence generation (57% to 36%), and blinding of outcome assessment (58% to 52%). Trial registration (37% to 47%) and the use of the CONSORT Statement (1%

**Data Availability Statement:** The risk of bias characterization was done with a large-batch-

customized-customized Python scripts (version 3; https://github.com/wmotte/robotreviewer_prob). The data management and analyses used R (version 3.6.1). All data including code and risk of bias data are available at https://github.com/wmotte/RCTQuality).

**Funding:** This work was supported by The Netherlands Organisation for Health Research and Development (ZonMw) grant "Fostering Responsible Research Practices" (445001002) (CV,JT,WO). The funders had no role in study design, data collection and analysis, decision to publish, or preparation of the manuscript.

**Competing interests:** The authors have declared that no competing interests exist.

**Abbreviations:** CDSR, Cochrane Database of Systematic Reviews; H-index, Hirsch index; ICMJE, International Committee of Medical Journal Editors; JIF, journal impact factor; PDF, Portable Document Format; RCT, randomized controlled trial; WHO, World Health Organization.

to 20%) also rapidly increased. In journals with a higher impact factor (>10), the risk of bias was consistently lower with higher levels of RCT registration and the use of the CONSORT Statement. Automated risk-of-bias predictions had accuracies above 70% for allocation concealment (70.7%), random sequence generation (72.1%), and blinding of patients/personnel (79.8%), but not for blinding of outcome assessment (62.7%). In conclusion, the likelihood of bias in RCTs has generally decreased over the last decades. This optimistic trend may be driven by increased knowledge augmented by mandatory trial registration and more stringent reporting guidelines and journal requirements. Nevertheless, relatively high probabilities of bias remain, particularly in journals with lower impact factors. This emphasizes that further improvement of RCT registration, conduct, and reporting is still urgently needed.

## Introduction

Randomized controlled trials (RCTs) are the primary source for evidence on the efficacy and safety of clinical interventions, and systematic reviews and clinical guidelines synthesize their results. Unfortunately, many RCTs have methodological flaws, and results are often biased [1]. Across RCTs, there is a major risk for inflated estimates and problems with randomization, allocation concealment, and blinding [2,3]. Recently, it was shown that over 40% of RCTs were at high risk of bias which could have been easily avoided [4,5]. Moreover, poor reporting prevents the adequate assessment of the methodological quality of RCTs and limits their reproducibility [6]. Avoidable sources of waste and inefficiency in clinical research were estimated to be as high as 85% [7].

Already in 1996, CONSORT criteria have been introduced to improve RCT reporting [8]. Moreover, mandatory RCT registration by the International Committee of Medical Journal Editors (ICMJE) has been put forward [9,10], with detailed registration before commencing the RCTs enabling more transparent and complete reporting. More recently, the importance of increasing value and reducing waste in medical research was emphasized, and meaningful steps were proposed toward more high-quality research, including improved methodology and reporting and reduction of unpublished negative findings [6,11]. Additional actions to improve methodological quality and transparency of RCTs include trial tracker initiatives aimed at reducing non-publication of clinical trials [12] and fostering responsible research practices. At the most recent World Conference on Research Integrity, the Hong Kong Principles were proposed for responsible research practices, transparent reporting, open science, valuing research diversity, and recognizing contributions to research and scholarly activity [13].

Even though these actions and initiatives have undoubtedly contributed to the awareness that the methodological quality of RCTs needs to improve, the question remains whether real progress has been made in reducing the extent of avoidable waste in clinical research. In other words, have these initiatives and measures improved the quality, transparency, and reproducibility of RCTs? Several studies have assessed the methodological quality of reporting and risk of bias in RCTs [14], but most are relatively small and limited to specific medical disciplines or periods. Nevertheless, based on 20,920 RCTs from Cochrane reviews published between 2011 and 2014, there are indications that poor reporting and inadequate methods have decreased over time [15]. However, large-scale evidence on trends of RCT characteristics and methodological quality across medical disciplines over time is currently lacking. This is surprising given the importance of valid and reliable evidence from RCTs for patient care. Therefore, this study aimed to provide a comprehensive analysis of developments in the clinical trial landscape

between 1966 and 2018 based on 176,620 full-text RCT publications. Specifically, we identified full-text RCTs via PubMed. We then used automated analyses to assess the risk of bias, CONSORT Statement reporting, trial registration, and characteristics related to publication (number of authors, journal impact factor [JIF], and medical discipline).

## Methods

A protocol for a prediction paper was registered before study conduct [16]. This protocol does not apply to the current manuscript, which is a description of the data set used in the protocol. Nevertheless, it does contain details on the methodology that we used in this manuscript. The database and scripts are available through GitHub (see Data sharing), and the results are disseminated through the medRxiv preprint server.

### Selection of RCTs and extraction of characteristics

RCTs were identified (November 20, 2017) via PubMed starting with all publications indicated as "randomized controlled trial" using the query "randomized controlled trial[pt] NOT (animals[mh] NOT humans[mh])." The initial search did not include a time window. Non-English language, nonrandomized, animal, pilot, and feasibility studies were subsequently excluded (see **S1 Text** for details on selection procedure). We collected the Portable Document Format (PDF) for all available RCTs across publishers in journals covered by the library subscription of our institution and converted these PDFs to structured text in XML format using publicly available software (Grobid, available via GitHub). By linking information from PubMed, the full-text publication, and data from Scopus and Web of Science, we extracted metadata on the number of authors, author gender, number of countries and institutions of (co-)authors, and the Hirsch (H)-index of the first and last authors (see **S1 Table** for details). Moreover, we extracted the JIF at the time of publication. Time was stratified in 5-year periods as behavioral changes are expected to occur at a relatively low pace, with the relatively few trials published before 1990 merged in one stratum.

### Risk-of-bias assessment

For every included full-text RCT, the risk-of-bias assessment was automatically performed using a machine learning assessment developed by RobotReviewer [17]. This tool is optimized for large-scale characterizations [18,19] and algorithmically based on a large sample of human-rated risk-of-bias reports and extracted support texts from trial publications covering the full RCT spectrum. RobotReviewer and human raters' level of agreement was similar for most domains (human–human agreement: min/max, 71% to 85%, average, 79%; human–RobotReviewer agreement: min/max, 39% to 91%, average 65%) [18,19]. Of the 7 risk-of-bias domains described by Cochrane [20], we assessed 4: random sequence generation and allocation concealment (i.e., selection bias), blinding of participants and personnel (i.e., performance bias), and blinding of outcome assessment (i.e., detection bias). Publication bias and outcome reporting bias were outside the scope of our analysis. The machine learning output yield a probability with values below 0.5 corresponding with "low risk" and values above 0.5 with "high or unclear" risk of bias.

### Analysis of trial registration and CONSORT Statement reporting

To check for trial registration, we extracted trial registration numbers from the abstract and full-text publication. We searched for the corresponding trial registration number in 2 online databases: the World Health Organization's (WHO) International Clinical Trials Registry

Platform, composed of worldwide primary and partner registries, and the ClinicalTrials.gov trial registry [21,22]. We checked all full-text publications for at least 1 mention of the words "Consolidated Standards of Reporting Trials" or CONSORT.

## Analysis related to journal impact factor

Even though the JIF (the average number of times its articles has been cited in other articles for 2 years) is not a very suitable indicator of journal quality [23], no unbiased alternatives exist. Therefore, in our study, we used the JIF as a proxy to identify journals with high publication standards and high rejection rates. For each trial, we selected the JIF of the year before trial publication. We used a JIF threshold of 10 as the primary cutoff based on JIF distributions (see **S2 Table** and **S1 Fig**) and previous evidence for sensitivity to assess RCT quality using this cutoff [15]. However, we also performed sensitivity analyses for JIF cutoff thresholds at 3 and 5.

## Analyses related to medical disciplines

We assigned RCTs to medical disciplines based on the journal category (Web of Science) [9]. As a secondary analysis, we examined medical disciplines separately. Medical disciplines with less than 4,000 RCTs in our sample were assigned to the category "Other."

## Power calculation and statistics

No a priori formal power calculation was performed as the aim of this project was to include all RCTs available on PubMed. Temporal patterns in the individual risk-of-bias predictions were modeled with regression analysis. Reported *P* values correspond to the overall trend estimate or the comparison of the average value per year in the 1990 to 1995 and 2010 to 2018 strata obtained from post hoc Tukey-corrected comparisons. This post-1990 period was chosen to cover the first years following significant awareness on the need to report transparently, in comparison to the latest years in our data set. Temporal patterns in trial registration and CONSORT Statement reporting were modeled with logistic regression. Because median values were very close to mean values, the data are presented as mean ± 95% confidence intervals.

## Risk-of-bias assessment validation

To determine the accuracy of the automated risk-of-bias assignment to RCTs in our database, we validated a large subset of RCTs with human assessments obtained from systematic reviews. To this end, we inspected all PMIDs of RCTs included in the 8,075 systematic reviews (Issue 6, 2020) published in the Cochrane Database of Systematic Reviews (CDSR). The CDSR is part of the international charitable Cochrane organization, with more than 50 review groups based at research institutions worldwide, aiming to facilitate evidence-based choices about health interventions. Contributions to the database come from these review groups as well as from ad hoc teams. The systematic reviews in the CDSR were identified through PubMed, for the period to 2000 and 2020. This was done with the NCBI's EUtils API with the following query: "Cochrane Database Syst Rev"[journal] AND "("2000/01/01"[PDAT]: "2020/05/31"[PDAT])." We limited the assessment of the latest systematic review updates to prevent overlap between included RCTs. Review protocols were excluded.

All systematic review tables containing the words "bias" and "risk" were inspected on human export risk-of-bias text associated with a PMID. If a PMID matched with a PMID in our full-text database and the risk-of-bias domain concerned "allocation concealment,"

"random sequence generation," "blinding of participants and personnel," or "blinding of the outcome," the human risk-of-bias text was extracted.

All human risk-of-bias texts and assigned judgments were manually inspected and assigned to the proper risk-of-bias category. Judgments were binarized into "low" and "high/unclear" risk of bias and used to validate our automated binarized risk-of-bias probabilities (i.e., $< =$ 0.5 "low" risk and $>$0.5 "high/unclear" risk) in terms of accuracy, sensitivity, specificity, and kappa. The accuracy is expressed as the proportion of those RCTs correctly categorized by the model, namely as (true positives + true negatives) or (true positives + false positives + false negatives + true negatives).

## Results

### RCT full-text acquisition process

From the 445,159 PubMed entries for RCTs, we identified 306,737 eligible RCTs (see flowchart in **Fig 1**). Full-text articles were obtained from 183,927 RCTs. RCT publications with an uncertain year of publication (7,307) were excluded, resulting in a final sample size of 176,620 RCTs. The distribution of the risk-of-bias domain probabilities of included and excluded RCTs was comparable, with very similar interquartile ranges (**S3 Table**). The presence of the "With CONSORT Statement" and "RCT Registration" outcomes were 3% to 4% lower in the excluded RCTs (**S4 Table**). This full-text sample size is in line with the overall number of potential RCTs in the PubMed database (**S2 Fig**).

### RCT characteristics over time

RCTs for which full texts were obtained were predominantly published over the last 3 annual strata with 89,373 publications in 2010 to 2018 (11,172 per year) compared to 6,066 publications between 1990 and 1995 (1,213 per year; **Fig 2A**). Over time, the average number of authors steadily increased from 5.2 (CI: 5.12 to 5.26) in 1990 to 1995 to 7.8 in 2010 to 2018 (CI: 7.76 to 7.83; $P < 0.0001$ for post hoc category difference) (**Fig 2B**). This was accompanied by a steady increase in the number of involved countries and institutions affiliated with all authors (number of institutions in 1990 to 1995: 3.24 (CI: 3.17 to 3.31) versus 2010 to 2018: 4.84 (CI: 4.81 to 4.87; $P < 0.0001$ for post hoc category difference)) (**Fig 2C and 2D**). In additional analyses, the percentage of female authors in RCTs has gradually increased over time, as well as the H-index of the first and last author (**S3 Fig**).

### Risk of bias, registration, and reporting: Trends over time

We found an overall continuous reduction in the risk of bias due to inadequate allocation concealment, dropping from 62.8% (CI: 62.4% to 63.1%) in trials published in 1990 to 1995 to 50.9% (CI: 50.7% to 51.0%; $P < 0.0001$ for overall trend) in trials published between 2010 and 2018 (**Fig 3A**). There was a relatively stronger decrease in the risk of bias due to nonrandom sequence generation, from 54.0% (CI: 53.6% to 54.5%) for trials in 1990 to 1995 to 36.4% (36.3% to 36.6%; $P < 0.001$ for overall trend) for trials in 2010 to 2018 (**Fig 3B**). The risk of bias due to not blinding participants and personnel showed a distinctly different pattern, with a sequential increase since 2000 up to 56.9 (56.8% to 57.1%; $P < 0.001$ for overall trend) in 2010 to 2018, after an initial decrease (**Fig 3C**). The risk of bias due to not blinding outcome assessment decreased over time, from 56.6 (56.4% to 56.9%) to 51.8 (51.7% to 51.9%; $P < 0.001$ for overall trend; **Fig 3D**). In all RCTs, mention of a trial registration number rapidly increased from close to 0 (0.37% for 1990 to 1995, $n = 6,272$) to up to 46.7% (46.4% to 47.0%; $n = 89,373$) in 2010 to 2018 (**Fig 3E**). We found very low reporting (1.05 (CI: 0.82% to 1.35%)

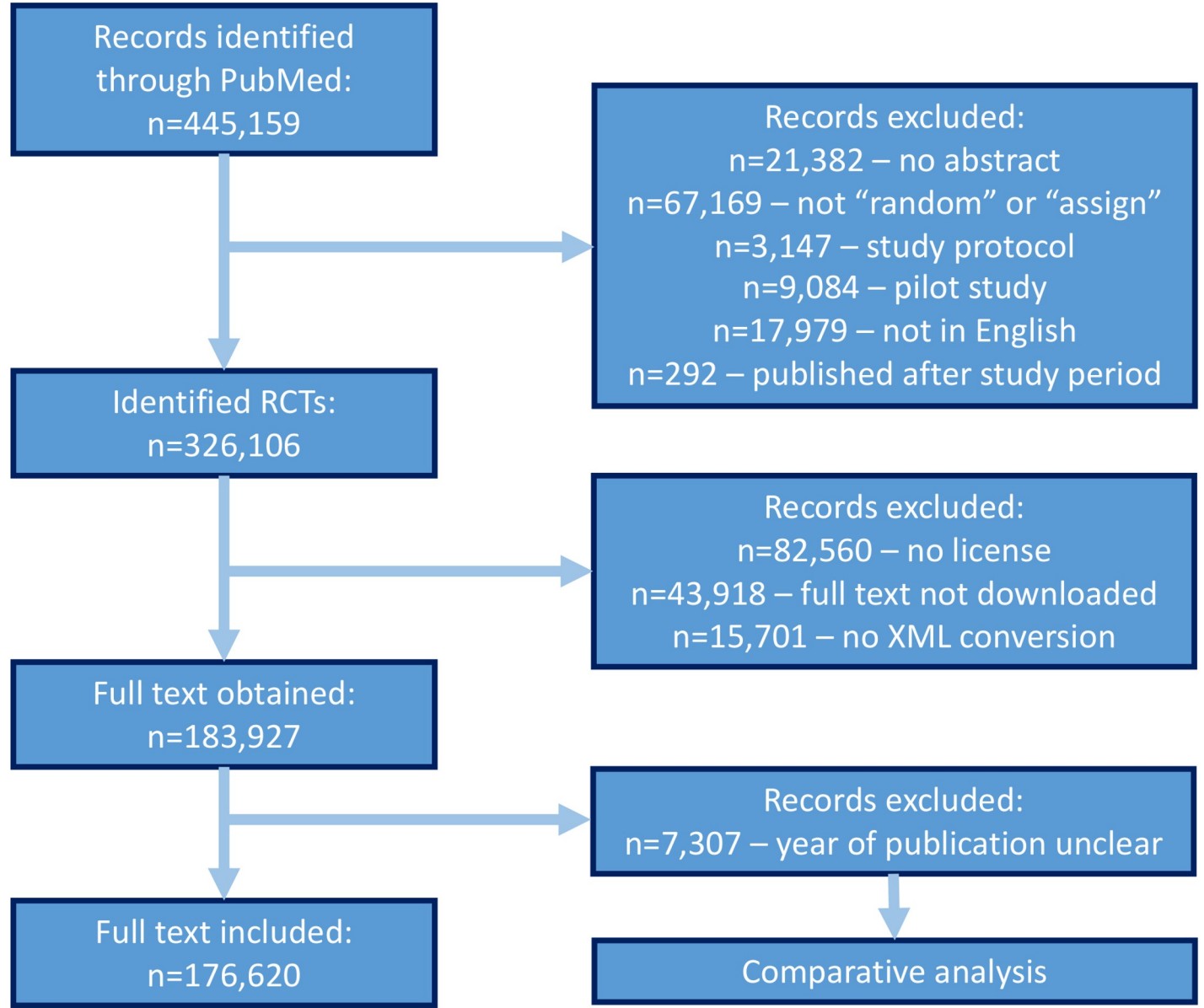

**Fig 1. Flowchart of how the final full-text randomized clinical trials were obtained. 1.** RCT, randomized controlled trial.

in 1990 to 1995, $n = 6,272$) of the CONSORT Statement in full-text RCT publications, contrasting with 19.5% (19.3% to 19.8%; $P < 0.0001$ for post hoc category difference; $n = 89,373$) of all trials between 2010 and 2018 (**Fig 3F**).

## Risk of bias and reporting: Relation with journal impact factor

The proportion of RCT publications with a lower risk of bias in allocation concealment was consistently lower in journals with JIF larger than 10 ($P < 0.001$; $P < 0.001$ for overall trend, **Fig 4A**). This also applied to randomization and blinding of participants and personnel and outcome assessment, even though the results were less pronounced compared to allocation concealment bias ($P < 0.0001$, all domains, for latest time point; **Fig 4B–4D**). For allocation bias, randomization bias, and blinding of outcome bias, the overall trends showed a decrease

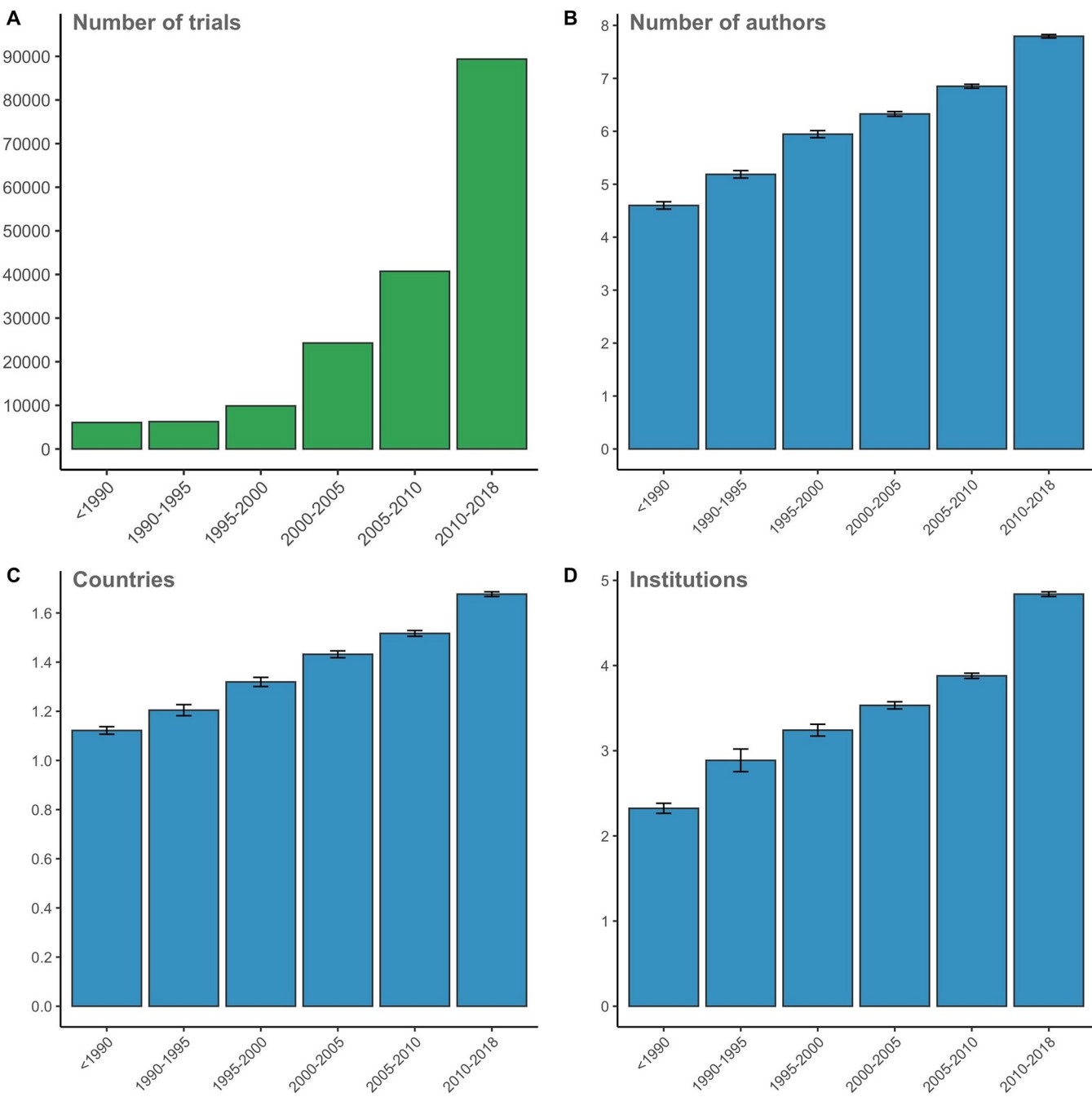

**Fig 2.** Number of RCTs included over time **(A)** and the corresponding average number of authors **(B)**, the number of countries involved **(C)**, and the number of institutions involved **(D)**. The indicated stratum range is up to but not including the last year. Raw data are available via DOI 10.5281/zenodo.4362238. RCT, randomized controlled trial.

over time ($P < 0.001$). Large differences were found in terms of trial registration and reporting of CONSORT between RCTs published in high or lower impact journals ($P < 0.0001$ for latest time point; $P < 0.001$ for overall trend, **Fig 4E and 4F**). Moreover, 73% (72% to 74%) of trials in journals with a JIF higher than 10 were registered, and 26% (25% to 27%) reported the CONSORT Statement between 2010 and 2018 (both measures: $P < 0.0001$ in comparison with JIF <10). Sensitivity analysis with JIF cutoff values of 3 and 5, respectively, yielded comparable

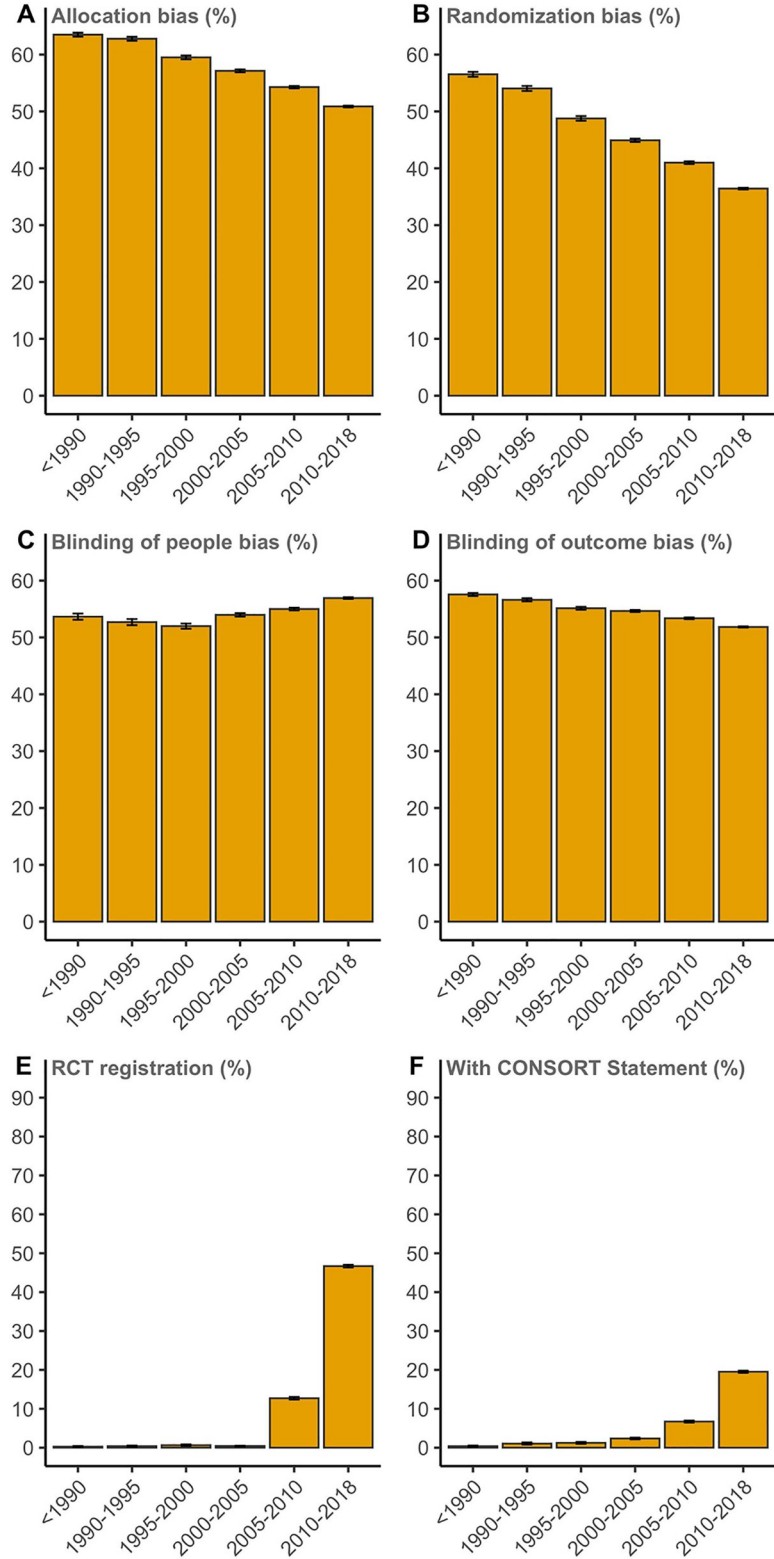

**Fig 3.** Risk of bias due to inadequate allocation concealment **(A)**, random sequence generation bias **(B)**, the bias in blinding of patients and personnel (people) **(C)**, the bias in blinding of outcome assessment **(D)**, RCT registration **(E)**, and reporting of the CONSORT Statement **(F)** for all RCTs plotted over time. The indicated stratum range is up to but not including the last year. Raw data are available via DOI 10.5281/zenodo.4362238. RCT, randomized controlled trial.

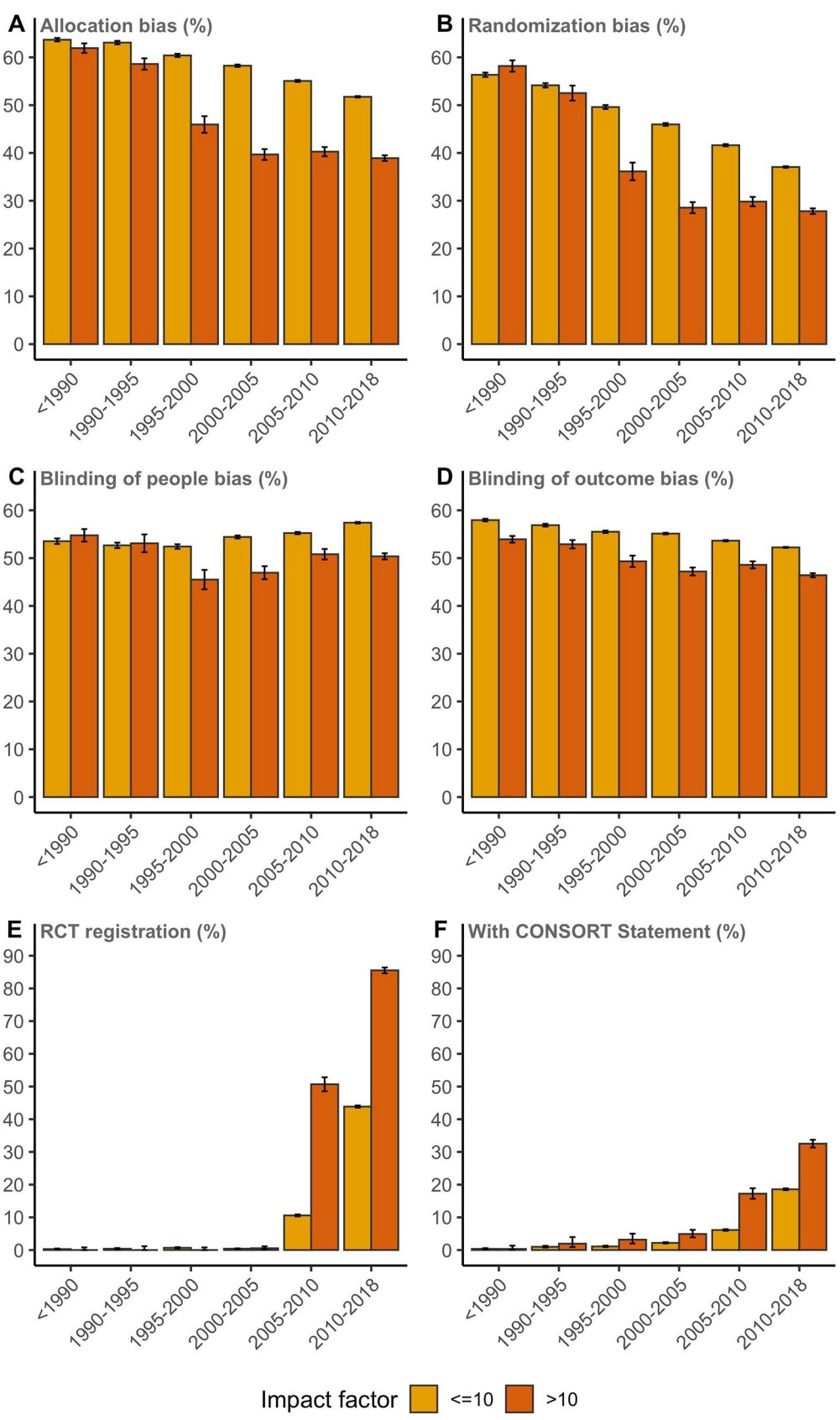

**Fig 4.** Risk of bias in allocation concealment (**A**), the bias in randomization (**B**), the bias in blinding of patients and personnel (people) (**C**), the bias in blinding of outcome assessment (**D**), RCT registration (**E**), and CONSORT Statement reporting (**F**) plotted over time for RCTs published in journals with JIF >10 and journals with JIF <10. The indicated stratum range is up to but not including the last year. Raw data are available via DOI 10.5281/zenodo. 4362238. JIF, journal impact factor; RCT, randomized controlled trial.

but smaller differences with reduced bias and increased registration and mentioning of the CONSORT Statement in journals with higher JIF (**S4 and S5 Figs**).

### Risk of bias and reporting: Relation with medical discipline

The risk-of-bias patterns substantially differed across medical disciplines (**S5 Table**). The lowest probabilities of bias were found in RCTs within the field of anesthesiology (27% randomization bias, 43% allocation concealment bias, 45% risk of bias due to insufficient blinding of participants and personnel, and 45% bias in blinding of outcome assessment) (**S6 Fig**). The field of oncology had the highest levels of trial registration (43.4%) and mention of the CONSORT Statement (30.3%) (**S7 Fig**). Registration rates were lowest in the field of endocrinology and metabolism (8.0%) and urology and nephrology (10.2%).

### Risk-of-bias assessment validation

In total, 63,327 matching human risk-of-bias judgments and automated risk-of-bias predictions were extracted from 17,394 unique RCTs included in a Cochrane systematic review (**Table 1**). Overall, automated accuracy in determining "randomization bias," "allocation bias," and "blinding of people bias" was above 70%. The "blinding of outcome bias" had a lower accuracy, namely 63.3%. The distribution of risk-of-bias predictions is shown in **S8 Fig**.

We also classified the chance-corrected level of agreement in terms of kappa. Kappa ranges from "poor" (<0.00), "slight" (0.00 to 0.20), "fair" (0.21 to 0.40), "moderate" (0.41 to 0.60), "substantial" (0.61 to 0.80) to "almost perfect" (0.81 to 1.0) [24]. Within this scale, a moderate agreement was found for randomization and blinding of people bias, and a fair agreement was found for allocation bias and blinding of outcome bias (**Table 1**).

### Discussion

We analyzed a total of 176,620 full-text publications of RCTs between 1966 and 2018 and show that the landscape of RCTs has considerably changed over time. The likelihood of bias in RCTs has generally decreased over the last decades. This optimistic trend may be driven by

**Table 1. Validation of the 4 risk-of-bias domains between automated and human assessments in RCTs obtained from the Cochrane systematic reviews risk-of-bias tables.**

| Domain | RCTs | Judgment | K | Accuracy (95% CI) | Sensitivity (95% CI) | Specificity (95% CI) | Kappa (95% CI) |
|---|---|---|---|---|---|---|---|
| Random sequence generation | 15,799 | High-Unclear | 5,840 | 72.1 (CI: 71.4–72.8) | 63.5 (CI: 62.3–64.8) | 76.5 (CI: 75.7–77.3) | 39.1 (CI: 37.5–40.6) |
| | | Low | 9,959 | | | | |
| Allocation concealment | 19,058 | High-Unclear | 9,672 | 70.7 (CI: 70.1–71.4) | 69.3 (CI: 68.4–70.2) | 72.5 (CI: 71.5–73.4) | 41.3 (CI: 40.0–42.6) |
| | | Low | 9,386 | | | | |
| Blinding of participants and personnel | 2,121 | High-Unclear | 1,400 | 74.8 (CI: 72.9–76.6) | 79.8 (CI: 77.8–81.9) | 63.8 (CI: 60.2–67.5) | 42.8 (CI: 38.7–47.0) |
| | | Low | 721 | | | | |
| Blinding of outcome assessment | 26,349 | High-Unclear | 14,009 | 62.7 (CI: 62.1–63.3) | 63.3 (CI: 62.6–64.1) | 61.7 (CI: 60.8–62.6) | 24.5 (CI: 23.3–25.6) |
| | | Low | 12,340 | | | | |

RCT, randomized controlled trial.

increased knowledge augmented by mandatory trial registration and more stringent reporting guidelines and journal requirements. Nevertheless, relatively high probabilities of bias remain, particularly in journals with lower impact factors.

Regarding the risk of bias, the trends that emerge from our analyses are certainly hopeful. The risk of bias in RCTs declined over the past decades, with lowering trends for bias related to random sequence generation, allocation concealment, and blinding of outcome assessment. In accordance, there is an increasing percentage of RCTs that are registered in public trial registers and use CONSORT guidelines. Despite 2 decades of documentation and calls for trial registration, it only substantially increased around 2004 when trial registration was made a condition for publication by the ICMJE. This policy was implemented and supported by WHO in July 2005 [25]. Our results are in line with the assessment of 20,920 RCTs from Cochrane reviews in 2017 that found improvements in reporting and methods over time for sequence generation and allocation concealment [15].

Notwithstanding these improvements, it is also clear that there is still a pressing need to further RCTs' methodological quality. The average risk in each of the bias domains remains generally high (around 50%), and bias related to blinding of participants and personnel increases over time, which may be due to more pragmatic or nondrug RCTs being performed. Moreover, despite the requirement of trial registration for publication since 2004, still in 2017 a substantial percentage of published RCTs do not report registration numbers. This suggests a lack of registration in a subset of RCT, although the absence of registration numbers does not necessarily imply the absence of registration. Furthermore, many RCTs do not mention the CONSORT guidelines in their full text, and more so for journals with lower impact factors. Despite the accessibility of reporting guidelines, researchers are generally not required to adhere to them. More problematic, requirements are not strictly enforced, and noncompliance to all the items on the reporting guideline is not sanctioned [4,26]. It is important to note that not mentioning CONSORT does not necessarily imply nontransparent reporting, although explicit mentioning is preferable. To further improve RCTs' methodological quality and reliability, there is still a long way to go. The rather slow progress of improvement may be due to the complex nature of conducting RCTs. Better education, enforcements, and (dis)incentives may be inevitable. Additionally, making data sets available according to the FAIR principles arguably will improve the situation [27].

Dependent on expectations and future goals, the interpretation of our findings can be either optimistic or pessimistic: Optimistic because, over the past decades, there has been quite some improvement in RCT conduct and reporting, but pessimistic because the improvements are going at a rather slow pace. From our analyses, it also appears that journals with higher JIF generally publish RCTs with lower predictions on the risk-of-bias domains. Our results confirm previous results showing higher JIF (higher than 10) being associated with a lower proportion of trials at unclear or high risk of bias in Cochrane reviews [15]. Even though JIFs are not a very suitable measure of journal quality, our results are in line with previous studies showing that increased JIF is related to a higher methodological quality of RCTs [28]. Finally, there are large differences across medical disciplines related to the risk-of-bias predictions across domains which may be related to the type and size of RCTs (i.e., more pragmatic RCTs) across medical disciplines.

With regard to authors, not only is there a growing number of authors and institutions from a growing number of countries involved in publications, but also a steadily increasing average H-index of the first and last author. The absolute and relative numbers of female authors in RCTs also gradually increased over time, with a large rise in first and last female authorships. Even though trends increase over time, the average percentage of female last authorships remains relatively low at 29% (2010 to 2018), in line with recent literature [29].

There are several strengths and limitations inherent to our approach of automated extraction of full-text RCT publications. The automated and uniform approach that yielded an unprecedentedly large and rich data source concerning RCTs from the last decades is available for further study (see https://github.com/wmotte/RCTQuality or DOI 10.5281/zenodo.4362238 for the data), covering a large proportion of all published RCTs included in PubMed. Moreover, we validated standardized human risk-of-bias judgments by trained reviewers from Cochrane systematic reviews with automated risk-of-bias predictions (Table 1). Nevertheless, there are several limitations. First, our selection of RCTs may have been biased as full-text RCTs may not have been missing at random (e.g., with the particular year of publication or RCT methodology). Most RCTs are not included as we do not have access to full-text data, making it difficult to make definitive conclusions on the missing RCTs concerning the risk of bias, CONSORT Statement reporting, and trial registration. However, an additional analysis of over 7,000 RCTs with an uncertain year of publication (7,307) that were excluded yielded a similar distribution of the risk-of-bias domain probabilities.

Moreover, the included RCTs in our analyses were generally in line with the overall number of potential RCTs in the PubMed database (**S6 Fig**), except for the period between 1993 and 1998 when libraries switched from scanned versions to online subscriptions, potentially lowering the yield of automated full-text downloads. However, this period is before the release and implementation of the discussed guidelines, and our sample still maps the important patterns in the overall RCT publications over time. Second, the risk of bias is inherently difficult to assess reliably. Experts' assessments of trials show that labeling the same trials for different Cochrane reviews resulted in substantial differences [30,31]. Probabilities assigned with machine learning are based on a large set of human-assigned labels, and a direct comparison shows computerized assessment performance of 71.0% agreement [32]. Our manual validation based on 63,327 extracted and standardized human risk-of-bias assessments, as published in a CDSR, showed accuracies of over 70% and kappa values between 0.25 and 0.43. Even though this sounds far from impressive, it should be borne in mind that agreement between human reviewers is often not much higher. Classification of risk-of-bias domains is a difficult task, both for humans and automated software. For example, based on 376 RCTs, the overall kappa values of interrater risk-of-bias predictions ranged from 0.40 to 0.42, although some domains were clearly higher, such as random sequence bias agreements between human raters (kappa 95% CI: 0.56 to 0.71) [30]. The low agreement between human raters is precisely why the risk-of-bias tool was recently revised into version 2 [33]. The relatively high variability and imperfect accuracy are particularly problematic for individual trial characterization, where the inclusion or exclusion of an individual trial due to incorrect risk assessment will have a large impact. In our study, however, we applied the risk-of-bias characterization differently and did not focus on individual trials but rather studied patterns in risk-of-bias distributions. Third, we did not investigate all aspects of methodological rigor. In our study, we did not check for forms of attrition bias (e.g., incomplete outcome data) or reporting bias (e.g., selective outcome reporting). Fourth, even though the CONSORT Statement was introduced to improve RCT reporting [8], the rapid increase of RCTs that mention following the CONSORT guideline does not guarantee adherence, and reporting methodological quality can remain suboptimal [14]. We were not able to automatically correct for the conventional and non-abbreviated use of the word "consort." This may have slightly increased our CONSORT Statement percentages and explains the very low but nonzero values in the earliest stratum.

Our comprehensive picture of the methodological quality of RCTs provides quantitative insight into the current state and trends over time. With many thousands of RCTs being published each year and thousands of clinical trials currently recruiting patients, this can help us

to understand the current situation better but also to find solutions for further improvement. These could include a more stringent adoption of measures to enforce transparent and credible trial publication, but also fine-tuning of stricter registration regulations. In conclusion, our comprehensive analyses of a large body of full-text RCTs show a slow and gradual improvement of methodological quality over the last decades. While RCTs certainly face challenges about their quality and reproducibility, and there is still ample room for improvement, our study is a first step in showing that all efforts that have been made to improve RCT practices may be paying off.

## Supporting information

**S1 Text. Detailed data collection procedures.**
(DOCX)

**S1 Table. Operationalization of variables for RCTs, authors, institutions, and journals.**
RCT, randomized controlled trial.
(DOCX)

**S2 Table. All journals with JIF higher than 10 in the year preceding any of the individual publications in our data set.** JIF, journal impact factor.
(DOCX)

**S3 Table. Quantiles of estimated risk-of-bias domain probabilities for included and excluded RCTs.** RCT, randomized controlled trial.
(DOCX)

**S4 Table. Percentages for the CONSORT Statement and Registration outcomes for included and excluded RCTs.** RCT, randomized controlled trial.
(DOCX)

**S5 Table. Total number ($N$) of trials published in the period 2005–2018 in the different medical disciplines with the number (K) and corresponding proportion (percentage with 95% confidence interval) of trials with a risk-of-bias probability below 50% (i.e., "low risk").**
(DOCX)

**S1 Fig. Distribution of JIFs of analyzed RCTs with JIF cutoffs of 3, 5, and 10 (dotted lines).** The JIF of a journal in the year following the publication date of the RCT was used. Density represents the probability of a trial to belong to a given impact factor. JIF, journal impact factor; RCT, randomized controlled trial.
(TIFF)

**S2 Fig. The number of included RCTs against the total number of RCTs indexed in the PubMed database for the study period (1966–2018).** The year 1993 and 1998 are marked with vertical dashed lines. RCT, randomized controlled trial.
(TIFF)

**S3 Fig. Percentage of female authors and H-indices for the first and the last author, per period, for the included RCTs.** RCT, randomized controlled trial.
(TIFF)

**S4 Fig.** Risk of bias due to inadequate allocation concealment (A), random sequence generation bias (B), the bias in blinding of patients and personnel (people) (C), the bias in blinding of outcome assessment (D), RCT registration (E), and mentioning of the CONSORT Statement

(F) plotted over time for RCTs published in journals with JIF >3 and journals with JIF <3. The indicated stratum range is up to but not including the last year. JIF, journal impact factor; RCT, randomized controlled trial.
(TIFF)

**S5 Fig.** Risk of bias in allocation concealment (A), the bias in randomization (B), the bias in blinding of patients and personnel (people) (C), the bias in blinding of outcome assessment (D), RCT registration (E), and mentioning of the CONSORT Statement (F) plotted over time for RCTs published in journals with JIF >5 and journals with JIF <5. The indicated stratum range is up to but not including the last year. JIF, journal impact factor; RCT, randomized controlled trial.
(TIFF)

**S6 Fig. The average risk of biases for trials published in the period 2005–2018 in different medical disciplines.** "random": bias in randomization; "allocation": bias in allocation concealment; "blinding of people": bias in blinding of patients and personnel; "blinding outcome": bias in the blinding of outcome assessment.
(TIFF)

**S7 Fig. Presence of RCT registration and CONSORT Statement in trials published between 2005 and 2018.** RCT, randomized controlled trial.
(TIFF)

**S8 Fig. The machine learning risk-of-bias probabilities are plotted as density profiles against the human rater risk categories: "High-Unclear" and "Low" for 63,327 matching RCTs.** RCT, randomized controlled trial.
(TIFF)

## Author Contributions

**Conceptualization:** Christiaan H. Vinkers, Herm J. Lamberink, David Moher, Willem M. Otte.

**Data curation:** Christiaan H. Vinkers, Johanna A. Damen, Willem M. Otte.

**Formal analysis:** Christiaan H. Vinkers, Willem M. Otte.

**Funding acquisition:** Christiaan H. Vinkers, Joeri K. Tijdink.

**Investigation:** Christiaan H. Vinkers, Lex Bouter, Johanna A. Damen.

**Methodology:** Christiaan H. Vinkers, Herm J. Lamberink, Joeri K. Tijdink, Pauline Heus, David Moher, Johanna A. Damen, Lotty Hooft, Willem M. Otte.

**Project administration:** Christiaan H. Vinkers, Pauline Heus.

**Resources:** Pauline Heus, Johanna A. Damen.

**Software:** Herm J. Lamberink, Willem M. Otte.

**Supervision:** Christiaan H. Vinkers, Joeri K. Tijdink, Lex Bouter, Paul Glasziou, David Moher, Lotty Hooft, Willem M. Otte.

**Writing – original draft:** Willem M. Otte.

**Writing – review & editing:** Herm J. Lamberink, Joeri K. Tijdink, Pauline Heus, Lex Bouter, Paul Glasziou, David Moher, Johanna A. Damen, Lotty Hooft.

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
