## [Editor Report · Decision Letter 0]

26 Aug 2020

Dear Dr Vinkers, 

Thank you for submitting your manuscript entitled "Randomized controlled trial quality has improved over time but is still not good enough: an analysis of 176,620 randomized controlled trials published between 1966 and 2018" for consideration as a Meta-Research Article by PLOS Biology.

Your manuscript has now been evaluated by the PLOS Biology editorial staff as well as by an academic editor with relevant expertise and I am writing to let you know that we would like to send your submission out for external peer review.

Please re-submit your manuscript within two working days, i.e. by Aug 28 2020 11:59PM.

Kind regards,

Aaron

Aaron Nicholas Bruns, Ph.D.,

Associate Editor

PLOS Biology

---

## [Decision Letter · Decision Letter 1]

4 Nov 2020

Dear Dr Vinkers,

Thank you very much for submitting your manuscript "Randomized controlled trial quality has improved over time but is still not good enough: an analysis of 176,620 randomized controlled trials published between 1966 and 2018" for consideration as a Meta-Research Article at PLOS Biology. Your manuscript has been evaluated by the PLOS Biology editors, an Academic Editor with relevant expertise, and by four independent reviewers.

IMPORTANT: You’ll see that the reviewers are broadly positive about your study, but raise a number of issues, most of which can be addressed by textual revisions, but some of which (e.g. for reviewers #1 and #3) may need further analyses. In addition, the Academic Editor has provided some guidance which I have pasted into the foot of the email; in those comments, s/he suggests some analyses that might help address some of the reviewers' concerns, and notes some departures from the OSF protocol. These comments should also be attended to.

In light of the reviews (below), we will not be able to accept the current version of the manuscript, but we would welcome re-submission of a much-revised version that takes into account the reviewers' and Academic Editor's comments. We cannot make any decision about publication until we have seen the revised manuscript and your response to the reviewers' comments. Your revised manuscript is also likely to be sent for further evaluation by the reviewers.

We expect to receive your revised manuscript within 3 months. 

**IMPORTANT - SUBMITTING YOUR REVISION**

*Re-submission Checklist*

*Published Peer Review*

*PLOS Data Policy*

*Blot and Gel Data Policy*

Sincerely,

Roli Roberts

Associate Editor,

abruns@plos.org,

PLOS Biology

REVIEWERS' COMMENTS:

Reviewer's Responses to Questions

PLOS authors have the option to publish the peer review history of their article (what does this mean?). If published, this will include your full peer review and any attached files.

Reviewer #1: No

Reviewer #2: Yes: Prof. Rustam Al-Shahi Salman

Reviewer #3: No

Reviewer #4: No

Reviewer #1:

This manuscript aims to report an analysis of the evolution over time of publication characteristics, risk of bias and quality of reporting of randomized controlled trials (RCTs).

It is based on a very large sample of RCTS analyzed with the help of machine learning tools. This is, to date, the largest mapping of RCTs. The manuscript is overall clear and well written. The topic is interesting, although not completely original regarding the evolution of risk of bias over time. I was slightly surprised by the choice of Plos Biology to submit this manuscript. Although Plos Biology is very interested in transparency and good methodological standards, I am not sure this journal publishes a lot of RCTs.

Please find below my comments. I hope you will find them helpful.

Major comments

- The objective should be well-formulated in particular I would be more precise at the end of the introduction section.

- A key question is the validity of the automated tools used to evaluate characteristics (for example author gender) or risk of bias.

o We need more elements to be sure that these tools work well. Regarding risk of bias, the authors indicate an agreement between humans and Robot Reviewer ranging from 39% to 91%. I did not find that very similar to the agreement between humans and 39% is a low rate. Did the authors conduct some manual verifications? The authors report that they link their data with Cochrane data. A validation for risk of bias could be the comparison between risk of bias assessment obtained from the automated tool with corresponding risk of bias reported in Cochrane reviews. 

o Regarding risk of bias, it is not fully detailed how it works. It seems that the result is rendered as a probability but this is not clear enough. Is it a probability of being at high risk of bias for the domain from 0 to 100%? Usually, the terms score is not adapted to risk of bias.

o Did the authors exclude trials presenting some particularities for which risk of bias assessment needs to be adapted? For example cluster or cross-over trials?

- The authors evaluated random sequence generation, allocation concealment, blinding of patients and personnel and blinding of outcome assessment as key domains for risk of bias. I did not understand why they did not consider incomplete outcome data as it is another key domain. The authors seem to link this domain with publication bias (methods section, page 7, line 132) or the sentence is unclear? 

- It seems from the statistical analysis that the authors evaluated the statistical significance of the comparisons between 1990-1995 and 2010-2018. Why that? A trend test considering all strata may be more appropriate.

- I think there are (too) many results in this study. The authors should be sure to be consistent between objectives and results presented. For example, in the abstract, the objective does not mention risk of bias. Some results are less important. It is interesting to evaluate the trends in numbers of authors, institutions, gender but why the use of positive and negative words. The list of positive, negative and neutral words is probably not exhaustive.

- The term 'quality' is very criticized. I would refer to risk of bias or at a minimum methodological quality

- Some elements are not clear in particular the start date for this study. In the abstract, it seems 1966. But it appears from the method that there is only one stratum before 1990 because there were relatively few trials before 1990. Does it mean that this stratum includes trials from 1966 to 1990? In the flow chart, it rather seems that the authors excluded RCTs published before 1988. Why? It is not mentioned in the methods. Please clarify.

Minor comments

- In the abstract, introduction and discussion, the authors should be careful regarding the wording used. For example, in the abstract: "responsible research practices and reporting guidelines are increasingly important but whether these efforts have improved RCT quality is unknown. The same in Introduction section: "In other words, have these initiatives and measures improved the quality, transparency and reproducibility of RCTS. The objective of this article is to evaluate the evolution over time. Whether improvements over time are related to the development and use of reporting guidelines is another question that cannot be answered with this study design. 

- This is not the "use" of CONSORT, this is the "reporting" of CONSORT in text. I think that many RCTS may be compliant with the CONSORT without mentioning the name.

- Introduction:

o First sentence of the second paragraph, instead of "For a longer time period", I would give the date of first publication of the CONSORT. I would not have presented CONSORT and trial registration in the same sentence as this suggests a link between them.

- In the methods section, the authors report search of Medline, should be added "via PubMed" after to be consistent with other parts of the manuscript

- Results section

o When mentioning trial registration, it should be interesting to give the number of trial registration from 2005 and 2010. The same for CONSORT from 2000.

o Page 14, line 230: not clear to what refers the p-value reported here? 

Reviewer #2:

[identifies himself as Rustam Al-Shahi Salman]

I congratulate the authors for completing an enormous and impactful body of work. I have no suggestions for improvement.

Reviewer #3:

This is a meta-research study investigating the evolution of the quality of reporting and key methods of randomized controlled trials (RCTs) using meta-data from a huge number of published RCTs. To handle such an impressive number of articles (> 175,000), the author relied on the use of machine learning (ML) algorithms to extract relevant data. The sample size is a major asset of the present study compared to many other meta-research studies, that have been hampered by much lower sample sizes. But there are some limitations, too, that could be better taken into account or better emphasized.

Major comments

1. 122 810 RCTs were excluded for technical reasons (no institutional license, failing to download full text …), which represents almost 70% of the number included. In an RCT or an epidemiological study, it would have been considered poor methodology to exclude 41% of the sample, and this deserves discussion. I do not claim that the results of the study are not worthwhile or more biased than other meta-research studies that have often used additional eligibility criteria (e.g. by focusing on a more limited range of journals, for instance). But again, we expect some extreme rigor for such type of studies, and the choices leading to the analyzed sample size and their possible impact on the results should be thoroughly discussed.

2. Automatic extraction of information from meta-data is at the core of the study, and it would have been virtually impossible to check all this by hand. But the algorithms used are imperfect (not that I claim humans would be perfect in determining the risk of bias, for instance). Some years ago, I have participated in a project where we used genderize, for instance, and results are not perfect. Was there an attempt to manually check the automated results on a subsample? Otherwise some presentation of the performance of the algorithm and the possible impact on the results may be useful. For RoB, the agreement of the ML algorithm with humans is reported to be 71% (or 65%, different figures and different references being used in the methods and discussion, which is confusing). What impact could this have on the results? In particular, do the confidence intervals reported account for possible misclassification?

3. The number of publications has been increasing over years, and as a mechanical consequence, the citations also. The average impact factors of journals has also increase, as well as the proportion of journals indexed in the Journal Citation Reports with IF > 10, for instance. More precisely, 1.1% of all journals listed in JCR (SCIE) in 199 had IF > 10, 1.7% in 2009 and 3.0% in 2019. Those proportions may have been quite higher in the journals considered in the review (but it's only a guess). As a result, why not use a relative IF cut-off (e.g. top 10% or top 5%), instead of using a fixed IF cut-off (≤10 vs >10)? Also, the table S2 lists journals classified as with IF > 10 for all individual publications in the dataset, but for instance Leukemia's IF only reached 10 or more in 2012. Similarly, for Annals of Oncology it was in 2016. It is striking that no RCT would be included for these journals before those years. Perhaps another IF that the one of JCR (Clarivate Analytics) was sued, but this should be specified.

4. The average increase of citations may also explain why the h-index of authors also increased. It is therefore difficult to disentangle trends specific to RCTs to global trends of the whole medical literature. Asking for a specific answer to this point would be unfair to some extent, because it may require a large amount of work, but from a scientific point of view, the question of whether the h-index of authors increased more for RCTs than for other types of studies seems more interesting that simply describing the evolution of the h-index. That said, I wonder whether this any question about the evolution if the h-index of authors is really relevant and useful in the present study, which is not about the evolution of the citation network among authors of the biomedical literature.

5. Risk of bias is not a binary concept in Cochrane's evaluation. The RoB2 tool classifies risk of bias judgments as low risk of bias, some concerns, and high risk of bias. What was considered as "risk of bias" in the study, "high risk of bias"? More clarity would be welcome.

Additional comments

1. It is unclear in the abstract what characteristics of publication and authors are, and how they would relate to the main issue which is about methodological quality and bias.

2. I agree with problems of reports of RCTs with randomization, allocation concealment, and blinding but is difficult to ascertain what is the true treatment effect, in particular because this would need to formally define the target population. I would therefore recommend softening the assertion "the majority of RCT findings have inflated estimates" because this is not really supported by strong evidence.

3. The (expected) benefit of registration could be better explained, instead of simply noting it has been made mandatory by IMCJE.

4. I acknowledge that wording can become extremely tedious when describing results, but writing "The risk of bias in allocation concealment was consistently lower in trials published in journals with JIF larger than 10" implies that the risk of bias in allocation concealment in a quantity, when in fact what is lower is the proportion of trials with low (?) risk of bias.

5. The link of reference 15 (protocol) does not work, but the following one does: "https://osf.io/27f53/#!". Please consider revising.

Reviewer #4:

Thanks for giving me the opportunity to review this manuscript, which explores quality of RCTs published between 1966 and 2018. The authors used machine learning techniques to extract information on study characteristics, risk of bias, trial registration, CONSORT statements, H-indexes. They found that overall quality of RCTs improved over time (lower bias, more CONSORT reporting and more study registrations), but a considerable amount of RCTs still lacks basic quality characteristics.

Major issues:

1. Introduction, line 89-91: The authors state that there is already a study on RCT quality that included more than 20,000 RCTs from Cochrane reviews. This seems to be a high number of RCTs, but the authors further state that "large-scale evidence (…) is lacking". It would be helpful to know as to why reference 14 does not provide enough evidence on time trends (i.e. bias as only RCTs from Cochrane reviews were included).

2. Methods, line 128: The authors report on a human-RobotReviewer agreement of 65% on average, stating that this is similar to 79% human-human agreement. It would be helpful if the authors provided reasons as to why this difference is irrelevant or does not impact validity of results

3. Methods, line 160, 161: The authors state that regression analysis was used, but it would be nice if the authors further explained details of the regression model.

4. Methods, line 118-120: Why did the authors summarize all articles form 2010 to 2018, when in the methods it is stated that 5-year periods were considered (plus prior 1990 in one stratum)?

5. Results, line 171: Full-text was only available for 60% of identified RCTs. I would suggest that the authors elaborate on potential risks this imposes on the results.

6. Discussion, line 291, 292: "Additionally, making data sets available according to the FAIR principles arguably will improve the situation". It would be helpful if the authors gave reasons as to why data availability was not included in the assessment criteria for RCTs.

Minor issues:

1. Introduction, line 81: The authors mention the Hong Kong Principles without further elaboration. It would be helpful to readers if the authors provided a short (i.e. one sentence or a relative clause) description of these principles. 

2. Figure 1: The authors should report all numbers as "n=" (also the subcategories that were excluded).

3. Figure 4: Please check the caption (drop "plotted over time")

4. Discussion, lines 302, 303: The authors could provide reasons why one medical field had higher RCT quality than others (i.e. more pragmatic RCTs in one area than in the other).

COMMENTS FROM THE ACADEMIC EDITOR [lightly edited]:

After reading the comments made by our reviewers, we would like to invite you resubmit your paper in which you address these. In particular, I encourage you to expand the manuscript somewhat to address the issue of selection of input. This selection comes with the methodology and is thus is inevitable, but understanding its effect in more depth is also pivotal in understanding the results. Through your methods you have indeed "seen" more papers than is possible with the more traditional hand work. Some extra digging to understand how problematic this selection is, is very much welcome. Some ideas to provide this insight could be:

- get some insight into missing data: what years/location/ type of journal etc where these missing papers?

- perform some sensitivity analyses to get a feeling of the impact of this selection on the results: worst case/best case scenarios, or even more sophisticated methods are welcome or even preferred.

I truly hope that you will be able to provide some extra quantified insights and not just leave it to a academic discussion based on hypothetical. The reason for this is this issue is not only fundamental to this particular paper, but to the whole approach of using machines instead of hands/eyes/brains in meta-research on reporting of scientific results.

I applaud your efforts to make your research as open as possible - github/osf. When looking at the OSF preregistered protocol, some major elements deviate or are missing. Some examples that just caught my eye are

*

"prediction" as the main aim in protocol vs emphasis on "over time" in manuscript

*

"main difference between included and excluded trials will be described" in protocol (see comments earlier)

*

consort: mentioning consort vs using statreviewer

*

description of excluded data and imputation of missing data (in line with earlier comments).

We understand that plans and the final result do not always line up, but there has to be a substantial overlap and explanation of any non-overlap to make sense for the reader. We encourage you to add this information to the manuscript or its supplemental information. If the current ms is more a description of the dataset and the "prediction" element will be described in a separate paper, please explain, and remove the reference to the OSF protocol. On a similar note, I encourage you to highlight substantial additions or changes to your methods during this peer review as such in the new version of the manuscript. This way, there is a continuous line between the preregistered protocol on OSF and the actual reporting in this manuscript. It also shows readers/prospective peer reviewers the added value of the peer review process.

---

## [Decision Letter · Decision Letter 2]

27 Jan 2021

Dear Dr Vinkers,

Thank you for submitting your revised Meta-Research Article entitled "The methodological quality of randomized controlled trials has improved but is still not good enough: an analysis of 176,620 randomized controlled trials published between 1966 and 2018" for publication in PLOS Biology. I've now obtained advice from two of the original reviewers and have discussed their comments with the Academic Editor. 

Based on the reviews, we will probably accept this manuscript for publication, assuming that you will modify the manuscript to address the remaining points raised by the reviewers. Please also make sure to address the data and other policy-related requests noted at the end of this email.

IMPORTANT:

a) Please attend to the remaining requests from reviewer #1.

b) Please re-word your Abstract to fit the verbose style that we use at PLOS Biology (not the current structured format that is more typical of clinical journals).

c) The current title is rather cumbersome and repetitive. Please change it to something more appealing; we suggest: "Analysis of the methodological quality of 176,620 randomized controlled trials published between 1966 and 2018 reveals a positive trend but urgent need for improvement."

d) Please attend to my Data Policy requests below.

We expect to receive your revised manuscript within two weeks. Your revisions should address the specific points made by each reviewer. 

-  a cover letter that should detail your responses to any editorial requests, if applicable

*Published Peer Review History*

*Early Version*

Sincerely,

Roli Roberts

Senior Editor,

rroberts@plos.org,

PLOS Biology

DATA POLICY:

Regardless of the method selected, please ensure that you provide the individual numerical values that underlie the summary data displayed in the following figure panels as they are essential for readers to assess your analysis and to reproduce it: Figs 2ABCD, 3ABCDEF, 4ABCDEF, S1, S2, S3, S4ABCDEF, S5ABCDEF, S6, S7, S8ABCD. NOTE: the numerical data provided should include all replicates AND the way in which the plotted mean and errors were derived (it should not present only the mean/average values).

REVIEWERS' COMMENTS:

Reviewer #1:

I would like to thank the authors for having answered my comments and modified the manuscript.

I have some additional comments:

- Objective: I would remove the word "unbiased": how do the authors can be sure that their evaluation is unbiased. 

o In the abstract, the sentence is not really clear: "we mapped RoB trends over time in RCT publications including journal and author characteristics". The end of the sentence is not clear, journal and author characteristics are not part of RoB

- I really thank the authors for having conducted a validation with RoB assessment reported in Cochrane reviews.

o The interest of using Cochrane reviews that could be highlighted is that RoB is evaluated the same way in all Cochrane reviews using the RoB tool by trained reviewers.

o How do the authors define the parameter accuracy?

o Which RoB did they take for trials included in two Cochrane reviews? Did this situation happen?

o The Kappa values are very low and lower than the Kappa values between reviewers assessing risk of bias using the Risk of bias tool (there is an abundant literature on the topic). 

o On which human reports and on how many are based algorithms of RobotReviewer regarding risk of bias?

o I don't understand when the authors say that "RoB data reported in CDSR not imposed or standardized", it is not true. The RoB judgment is almost always expressed the same way in high, low or unclear (or yes, no, unclear) as it is the format used in Revman. It is possible to extract data from the Cochrane review as an xml file, and to directly extract the RoB table with the judgment. I don't know if this is what the authors did.

- In the statistical analysis, 

o I don't understand why the authors reported either p-values for trends or p-values for comparisons across two categories. It is not consistent throughout the manuscript. They reported trend tests for RoB but not for publication characteristics. Why? Trend tests could be used.

- In the results

o It is not clear from the abstract whether the percentages refer to the first and last time categories or not (first and last decades) and for the first time category, is it <1990 or 1990-1995?

o I am a bit lost with the time categories. Looking at the text, the oldest category seems to be 1990-1995, once <1990 is reported in text (why?) but looking at the Figures, <1990 is systematically reported. I think it would be better to be consistent throughout. Either the authors consider that trials published before 1990 are too old and they exclude them, or if not, they have to use the category <1990 as the reference.

- In the Discussion

o The summary of main findings puts a major emphasis on author characteristics including women while only a few results are presented on this part. This should be consistent.

o Many authors do not report registration number in trial publication but this does not mean that these trials were not registered (I agree that it is important to report the registration number in the publication). In contrast, it is not because you don't mention Consort in text that you do not comply with. Therefore I suggest the authors to be a little bit more careful in wording

o The term score is still mentioned

o To evaluate the item incomplete outcome data, it is not necessary to rely on trial registration. Access to trial registration is mainly important for selective outcome reporting.

- Table 1: I would present 95% CI everywhere and I would take the usual formulation for Rob items

Reviewer #4:

I would like to congratulate the authors for the improvement. All issues have been addressed adequately.

---

## [Editor Report · Decision Letter 3]

1 Mar 2021

Dear Christiaan,

On behalf of my colleagues and the Academic Editor, Bob Siegerink, I'm pleased to say that we can in principle offer to publish your Meta-Research Article "The methodological quality of 176,620 randomized controlled trials published between 1966 and 2018 reveals a positive trend but also urgent need for improvement" in PLOS Biology, provided you address any remaining formatting and reporting issues. These will be detailed in an email that will follow this letter and that you will usually receive within 2-3 business days, during which time no action is required from you. Please note that we will not be able to formally accept your manuscript and schedule it for publication until you have made the required changes.

PRESS: We frequently collaborate with press offices. If your institution or institutions have a press office, please notify them about your upcoming paper at this point, to enable them to help maximise its impact. If the press office is planning to promote your findings, we would be grateful if they could coordinate with biologypress@plos.org. If you have not yet opted out of the early version process, we ask that you notify us immediately of any press plans so that we may do so on your behalf.

Thank you again for supporting Open Access publishing. We look forward to publishing your paper in PLOS Biology. 

Sincerely, 

Roli

Roland G Roberts, PhD 

Senior Editor 

PLOS Biology